# Influence of Heat Stress and Water Availability on Productivity of Silage Maize (*Zea mays* L.) under Different Tillage and Fertilizer Management Practices in Rostov Region of Russia

Emma Gaevaya *, Izida Ilyinskaya, Olga Bezuglova, Alexander Klimenko, Sergei Taradin, Ekaterina Nezhinskaya, Anna Mishchenko and Andrey Gorovtsov

Federal State Budget Scientific Institution, Federal Rostov Agricultural Research Centre, Rassvet Village 346735, Aksay Region, Rostovskaja Oblast, Russia
* Correspondence: emmaksay@inbox.ru

**Abstract:** The aridization of the climate negatively affects the growth and development of plants and their productivity. The aim of this study was to determine the effect of heat stress and water availability on maize for silage under aridization and to find out the effectiveness of technological methods to maximize the use of available moisture. A long-term multifactorial experiment was performed during the period 1991–2020 in the region south of Rostov, Russia. The long-term multifactorial experiment is located in a zone of high heat supply and insufficient moisture, so the latter factor limits the yield of corn and dictates the need to find solutions to increase its environmental sustainability in extreme weather conditions. The values of the hydrothermal coefficient (HTC) were determined for the period of maize cultivation for each year of the study, ranked in ascending order and grouped into clusters. The results were mathematically processed by calculating the mean values (M) and their standard deviations (±SEM) with Statistica 13.3 software. The optimum conditions for silage maize development were found under the combination of 265 mm of rainfall and a 19.8 °C average temperature during the growing season. The maximum yield of silage maize was 33.8–45.2 t ha$^{-1}$. In dry years (HTC = 0.3), tillage had an advantage: the yield increase was 0.2–1.6 t ha$^{-1}$. In wet years, moldboard tillage was preferable: the yield increase was 0.3–2.9 t ha$^{-1}$. The application of farmyard manure for fertilization increased the yield by 10.5–41.9%. Increasing the fertilizer rate by 1.5 times increased the yield by 21.0–59.8%. In drought, tillage and average fertilizer rate provided returns by increasing yields up to 7.7 kg/kg. Our study provides valuable recommendations in fodder production, promotes moisture conservation, preserves soil fertility on the slopes when cultivating corn for silage, and will be useful to specialists in improving the efficiency of agricultural production.

**Keywords:** calcic chernozem (pachic); chisel moldboard; chisel tillage; fertilizers; heat stress; hydrothermal coefficient; yield



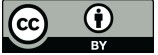

## 1. Introduction

Corn (*Zea mays* L.) belongs to the world's most important food, industrial and feed crop. According to the World Food and Agriculture Organization of the United Nations (FAO), more than 600 different basic products and by-products are produced from corn worldwide [1,2]. The yield of corn for silage in Russia ranges from 50–60 t ha$^{-1}$; under the arid conditions of the Rostov region, it decreases to 25–45 t ha$^{-1}$ [3].

In modern economic conditions, fertilizers are an expensive way to increase the corn yield [4,5]. The use of fertilizers is considered rational if the costs of their application are compensated by the yield [6]. To ensure the return on fertilizers used for corn, it is necessary to determine the application rates, taking into account the soil fertility, climatic conditions of the growing area, response and potential productivity, and real possible yield increase [7,8].

Contemporary agriculture has an urgent need to increase the efficiency of mineral fertilizers. Firstly, they should provide plants with mineral elements in optimal amounts at all stages of development. Secondly, mineral fertilizers should replenish the soil pool of the chemical elements used by the plants for crop production. Rich nutrition is needed to achieve high yield [9]. Depending on the meteorological characteristics of the growing season, the yield increase due to fertilizers varies widely [10].

The heat stress under the lack of precipitation in July–August in the corn blooming and filling period causes a reduction in yield. The application of nitrogen fertilizers during the soil drought makes it possible to obtain the highest increase of silage maize. With a favorable moisture regime, the highest return on fertilizers has been obtained as a result of the increased application rates [11].

In arid conditions, the method of tillage is of great importance. The main advantage of chisel cultivation is the reduction in plowing costs; the prevention of water and wind erosion on flat and sloping areas; the improvement of physical, mechanical and biochemical parameters; the preservation of moisture in soil horizons; as well as the increase in fertility. In erosion-prone agricultural landscapes, soil protection technologies are based on non-moldboard tillage, which reduces soil erosion by 2–4 times in crop rotations with tilled crops [12].

The weather as a component of the climate is always dynamic. Therefore, even a high level of agricultural technology intended for the average climatic conditions is not able to eliminate annual variations of yields. The wide spatiotemporal variability of natural factors, primarily agrometeorological conditions and soil fertility parameters, provides for a multivariate use of agricultural resource1 potential of the territory. The adaptation of agrotechnical solutions to the peculiarities of environmental conditions is one of the main reasons for the sustainable development of agriculture and an alternative to overcoming their adverse consequences [13].

Heat stress alone and in combination with drought is becoming another major constraint to maize production [14]. Temperatures are projected to increase by at least 1 °C, depending on the emissions scenario [15]. An increase in temperature of 2 °C would result in a greater reduction in maize yields than a decrease in precipitation of 20% [16,17].

The crop yield varies from year to year under the influence of meteorological factors, which makes it difficult to objectively assess the effectiveness of tillage methods [18]. This problem can be solved on the basis of systematic studies in long-term experiments, which are a theoretical information base for determining the effectiveness of agroclimatic, soil, and other factors [19].

This article presents the results of research on corn cultivation conditions on slope erosion-hazardous lands, which determines the novelty of the work and significantly expands the range of application of the results in practice.

The purpose of our study is to determine the influence of heat stress and water availability on the silage maize yield under different tillage and fertilization practices and to find out the effectiveness of technological methods that ensure the maximum use of available moisture.

## 2. Materials and Methods

### 2.1. Description of Study Site

These studies were carried out over the period 1991–2020 at the Federal State Budgetary Scientific Institution "Federal Rostov Agricultural Research Centre" in a long-term multifactorial experiment. Coordinates of the experimental plot: 47.370022 N, 39.909648 E; 47.36958 N, 39.90929 E. The experiment was registered in the Russian Geographical Network of long-term experiments with fertilizers application (certificate No. 169).

### 2.2. Experimental Design, Treatments and Cultural Practices

The experimental plot was located on the slope with southeastern exposure; with a steepness of up to 3.5–4.0°, erosion processes are weakly expressed. The soil of the

experimental plot is Calcic Chernozem (Pachic). The average annual runoff is about 20 mm. Horizon thickness is A—25–30 cm, A + B—from 30 to 60 cm depending on washout. The porosity of the arable horizon is 61.5%, of the subsoil—54%. The lowest moisture content is 34% and wilting moisture is 15.4%. The initial humus content in the arable layer of 0–30 cm at the beginning of the experiment was 3.8–3.83%, total nitrogen was 0.14–0.16%, mobile phosphates were 15.7–18.2 mg, and exchangeable potassium was 282–337 mg/kg of soil.

The experiment layout included three repetitions. The silage maize was studied in a five-field crop rotation: silage maize, winter wheat, spring barley, perennial grasses. The mid-season corn hybrid Zernogradsky 282 MV was sown, bred at the Donskoy Agricultural Research Center, Zernograd, Russia. The plots were arranged in a randomized manner. Released early-maturing hybrids were sown. An organomineral fertilizer system was applied: "0"—control plot without fertilizers; "1"—5 tons of farmyard manure (FYM) together with mineral fertilizers $N_{80}P_{60}K_{80}$ kg of active agent; "2"—8 tons FYM together with mineral fertilizers $N_{120}P_{90}K_{120}$ kg of active agent per 1 hectare of crop rotation area, as well as two tillage systems—chisel and moldboard. Phosphorus (ammophos) and potash (potassium chloride) fertilizers were applied before the main tillage; top dressing was carried out with ammonium nitrate in the phase of 2–4 leaves.

Sowing was carried out in 1–2 over ten days in May. The silage maize was mowed in 1–2 over ten days in August. The average growing season of silage maize is 105 days. The crop productivity of silage maize was studied in five-field crop rotation in three replicates and was determined using B.A. Dospekhov's method [20]. The plot area was 0.2 ha. Accounting for the crop was carried out with a declared area of 25 m$^2$, followed by weighing.

### 2.3. Data Observation and Calculation

The initial values of precipitation and temperatures were obtained at the meteorological station in Rassvet village using Precision Weather Station Vantage Pro2.

The climate of the research area is dry, moderately hot, and continental. The long-term average annual precipitation is 492 mm. During the spring–summer period, the rainfall is 260–300 mm. The average annual temperature is 8.8 °C, the average temperature in January is −6.6 °C, the minimum temperature in winter is −41 °C and in July +23 °C, and the maximum temperature in summer is up to +40 °C. The sum of active temperatures is 3210–3400 °C. Often, there are dry winds and dust storms of varying intensity [21] (Table 1).

**Table 1.** Long-term average annual values of meteorological conditions during the growing season of silage maize, Aksaysky district.

| Period | Precipitation, mm | Temperature, °C | Hydrothermal Coefficient (HTC) |
|--------|-------------------|-----------------|-------------------------------|
| May | 48 | 16.1 | 1.0 |
| June | 67 | 19.9 | 1.1 |
| July | 54 | 23.0 | 0.8 |
| Growing season | 169 | 19.7 | 0.9 |

The hydrothermal coefficient (HTC) of the corn growing season was calculated using G.T. Selyaninov's method [22] based on the actual values of the sum of precipitation and the sum of air temperatures. The calculation was made using the formula

$$K = \frac{10P}{\Sigma T};$$  (1)

where *P*—the amount of precipitation in millimeters for the period with temperatures above +10 °C;

$\Sigma T$—the sum of air temperatures in degrees Celsius (°C) for the same period.

The long-term average annual value of HTC was 0.9.

For each year of these studies, the HTC values of the corn growing season were determined. They were ranked in ascending order and grouped within the intervals of

the HTC values into the following clusters: "semi-dry" with the HTC 0.2–0.3; "very arid" 0.4–0.5; "arid" 0.6–0.8; "semi-arid" 0.7–1.0; "subhumid" 1.0–1.2; and "humid" 1.2–1.7 [20].

### 2.4. Statistical and Data Analysis

Two-way analysis of variance (ANOVA) was performed to assess the influence of tillage methods and fertilizer application rates on the productivity of silage maize. Fisher LSD test was used as a post-hoc test at the 95% level ($p \leq 0.05$) of significance. Statistical processing and visualization of the obtained results was carried out by variance analysis using Microsoft Excel 2011 and Statistica 13 (StatSoft, Tulsa, OK, USA) [23].

## 3. Results

### 3.1. Characteristics of the Meteorological Conditions for the Cultivation of Silage Maize

As a result of the summary of the long-term observations (30 years), it was found that the weather conditions of the growing season of silage maize over the years of observations differed significantly. The amount of precipitation in May, June, July, and the growing season as a whole was less than the long-term average annual values for "semi-dry cluster" by 20–80%, "very arid" by 45–58% and "arid" by 1.0–23%. In the same periods, the least amount of precipitation was recorded in May (1.4–5 times less than the average annual norm). In the "semi-arid cluster", the amount of precipitation was 4.8–36.2% higher than the average annual norm, in the "subhumid" by 14.6–45.6% and in the "humid" by 61.2–78.8% (Table 2).

**Table 2.** Characteristics of the meteorological conditions of the growing season of silage maize, 1991–2020.

| Parameter | Months | Long-term Average *n* = 30 | Clusters | | | | | |
|---|---|---|---|---|---|---|---|---|
| | | | Semi-Dry *n* = 5 | Very Arid *n* = 5 | Arid *n* = 5 | Semi-Arid *n* = 5 | Subhumid *n* = 5 | Humid *n* = 5 |
| Precipitation, amount, mm | May | 54.1 | 10.6 | 36.7 | 54.4 | 73.7 | 62.0 | 87.2 |
| | June | 55.7 | 44.2 | 35.2 | 42.8 | 40.8 | 81.2 | 90.2 |
| | July | 49.0 | 10.8 | 33.8 | 40.5 | 51.4 | 70.0 | 87.6 |
| | Growing season | 158.8 | 65.6 | 105.6 | 137.7 | 165.8 | 213.1 | 265.0 |
| Temperature, average, °C | May | 17.5 | 17.7 | 18.2 | 18.5 | 16.7 | 17.0 | 16.8 |
| | June | 21.8 | 21.9 | 22.8 | 22.0 | 23.4 | 20.8 | 20.1 |
| | July | 24.1 | 23.9 | 24.3 | 25.4 | 24.7 | 23.6 | 22.4 |
| | Growing season | 21.2 | 21.1 | 22.0 | 21.7 | 21.8 | 21.6 | 19.8 |
| Sum of air temperatures. °C | | 1945 | 1944 | 2020 | 1997 | 2007 | 1881 | 1820 |
| HTC | | 0.8 | 0.3 | 0.5 | 0.7 | 0.8 | 1.1 | 1.5 |

In the "semi-dry cluster" the amount of precipitation during the corn growing season was less than the long-term average value by 93.2 mm (58.7%). In the "very arid" cluster, the precipitation deficit was reduced up to 53.2 mm, and in the "arid" one up to 21.1 mm. Starting from the "semi-arid cluster", there was a tendency towards an increase in the precipitation amount during the growing season by 7.0 mm. A statistically significant increase in the precipitation is noted in the "subhumid cluster" (54.3 mm) and in the "humid cluster" (106.2 mm). The "humid cluster" was characterized by the highest amount of precipitation.

The average air temperature was more constant than the amount of precipitation. The actual average temperature values exceeded the long-term average annual values for the corresponding periods in the "semi-dry", "very arid" and "arid" clusters by a statistically insignificant value (0.1–1.4 °C; $p < 0.05$). The average monthly air temperature in the "semi-arid "and "subhumid clusters" was 0.2–1.0 °C lower than the long-term average annual values. In the "humid cluster", the average monthly temperatures in May and June were lower than the long-term average annual values in the same months by 0.7–1.7 °C ($p > 0.05$).

During the 30-year observation period, the HTC was 0.8. Its minimum values (0.3) are accounted for by the "semi-dry cluster". With an increase in precipitation and a decrease

in the sum of the temperatures, the HTC increased and reached its maximum (1.5) in the years of the "humid cluster".

### 3.2. Productivity of Silage Maize

The long-term average annual productivity of silage maize varied in the range from 19.7 to 27.6 t ha$^{-1}$ with an insignificant advantage with moldboard tillage. The productivity of silage maize with natural nutrition ranged from 12.8 to 13.1 t ha$^{-1}$ in the years of the "semi-dry cluster" and increased to 33.8—34.1 t ha$^{-1}$ in the years of the "humid cluster".

The application of an organomineral fertilizer system in the amount of 5 tons of FYM together with $N_{80}P_{60}K_{80}$, kg of active agent increased the yield by 4.3–4.5 t ha$^{-1}$ compared to the plot with natural fertility (21.5–26.9%). An increase in the fertilizer rate by 1.5 times (8 t FYM + $N_{120}P_{90}K_{120}$ kg ha$^{-1}$ of active agent) was accompanied by a 37.6–38.9% yield increase (Table 3).

**Table 3.** Productivity of silage maize depending on tillage method and the level of fertilization, t ha$^{-1}$,1991–2020.

| Clusters | Tillage Method | Fertilize Application Rate | | | Least Significant Difference ($p$ = 0.05). Tillage Fertilizer |
|---|---|---|---|---|---|
| | | «0» | «1» | «2» | |
| Semi-dry N = 5 | Chisel | 13.1 ± 2.7 | 15.3 ± 2.6 *** | 15.8 ± 2.5 ** | 0.55 |
| | Moldboard | 12.8 ± 2.3 | 14.7 ± 2.3 ** | 15.7 ± 2.3 *** | 0.75 |
| Very arid N = 5 | Chisel | 15.1 ± 0.6 | 19.8 ± 1.9 * | 23.4 ± 2.3 ** | 1.28 |
| | Moldboard | 14.7 ± 0.7 | 19.1 ± 1.8 ** | 22.8 ± 2.3 ** | 1.74 |
| Arid N = 5 | Chisel | 17.1 ± 1.2 | 20.1 ± 0.8 ** | 23.7 ± 1.3 *** | 0.81 |
| | Moldboard | 16.5 ± 1.2 | 20.0 ± 1.1 ** | 22.1 ± 1.4 *** | 1.11 |
| Semi-arid N = 5 | Chisel | 16.9 ± 4.7 | 21.2 ± 5.7 ** | 23.3 ± 5.9 *** | 1.12 |
| | Moldboard | 18.6 ± 4.8 | 21.7 ± 5.4 * | 23.8 ± 6.1 ** | 1.53 |
| Subhumid N = 5 | Chisel | 22.5 ± 1.9 | 31.9 ± 1.8 ** | 35.9 ± 2.0 ** | 1.60 |
| | Moldboard | 23.9 ± 2.0 | 32.1 ± 1.8 ** | 36.1 ± 1.9 ** | 2.18 |
| Humid N = 5 | Chisel | 33.8 ± 2.3 | 37.4 ± 0.8 * | 42.4 ± 2.1 * | 2.10 |
| | Moldboard | 34.1 ± 2.7 | 38.7 ± 1.0 * | 45.4 ± 1.8 ** | 2.68 |
| Long-term average annual values N = 30 | Chisel | 19.7 ± 1.6 | 24.3 ± 1.8 *** | 27.4 ± 2.0 *** | 0.71 |
| | Moldboard | 20.1 ± 1.7 | 24.4 ± 1.8 *** | 27.6 ± 2.2 *** | 0.69 |

Note: "0"—control plot without fertilizers; "1"—FYM 5 t + $N_{80}P_{60}K_{80}$; "2"—FYM 8 t + $N_{120}P_{90}K_{120}$. * $p < 0.05$, ** $p < 0.01$, *** $p < 0.001$.

The analysis of the yield data showed that in the "semi-dry cluster" the yield was lower than the long-term average annual yield by 33.6–43.3% and amounted to 12.8–15.8 t ha$^{-1}$. In the "very arid" and "arid cluster", the difference in the long-term average annual values was 14.1–22.7 t ha$^{-1}$ (13.5–23.5%). In other clusters, the silage maize differed slightly from the long-term average annual data. The exception was the "humid cluster", where the yield exceeded the long-term average annual values by 53.9–71.2%. The yield increase (14.0–17.6 t ha$^{-1}$) in this cluster was statistically significant.

In the clusters with HTC values from 0.3 ("semi-dry cluster") up to 0.5 ("very arid cluster"), the yield was lower than the long-term average by 4.0–11.9 t ha$^{-1}$. In the "arid cluster" (HTC = 0.7), the decrease was slighter—2.7–5.5 t ha$^{-1}$. In these years, the chisel tillage had an advantage. With the increase in water availability, the advantage of moldboard tillage begins to manifest: with the HTC = 0.9 in the "semi-arid cluster", the yield increase of silage maize was 1.5–4.1 t ha$^{-1}$; in the "subhumid cluster" (HTC = 1.2), the increase was 0.7–2.3 t ha$^{-1}$; and in the "humid cluster" it was 4.1–5.0 t ha$^{-1}$.

The increased temperature regime and insufficient precipitation are accompanied by the soil drying out, which leads to heat stress for plants. These unfavorable conditions influenced the silage maize. It varied significantly depending on the meteorological conditions of the growing season—the coefficient of variation of the long-term average annual values was 42.3–48.1% (Table 4).

**Table 4.** Coefficient of variation of silage maize yield under the influence of tillage and nutritional level, %, 1991–2020.

| Clusters | Tillage, Fertilizer Application Rate | | | | | |
| --- | --- | --- | --- | --- | --- | --- |
| | Chisel | | | Moldboard | | |
| | «0» | «1» | «2» | «0» | «1» | «2» |
| Semi-dry | 46.9 | 38.5 | 34.7 | 40.5 | 34.4 | 32.6 |
| Very arid | 9.1 | 21.3 | 21.5 | 10.1 | 20.5 | 21.7 |
| Arid | 15.3 | 8.4 | 12.5 | 15.7 | 12.7 | 14.3 |
| Semi-arid | 62.0 | 60.4 | 56.3 | 57.6 | 56.1 | 57.1 |
| Subhumid | 19.0 | 12.9 | 12.7 | 18.8 | 12.4 | 11.8 |
| Humid | 15.1 | 4.7 | 11.2 | 17.9 | 5.9 | 8.9 |
| Long-term average annual | 44.4 | 40.0 | 40.1 | 45.1 | 41.2 | 43.0 |

Note: "0"—control plot without fertilizers; "1"—FYM 5 t + $N_{80}P_{60}K_{80}$; "2"—FYM 8 t + $N_{120}P_{90}K_{120}$.

The variability of the silage maize yield in the "semi-dry cluster" was also significant—32.6–46.9%. As the heat stress decreased, the coefficient of variability reduced to 9.3–29.3%.

### 3.3. Factors Affecting the Silage Maize Yield

#### 3.3.1. Influence of Meteorological Conditions

To assess the impact of heat stress and water availability on maize productivity, depending on the tillage method and the fertilization level, a correlation-regression data analysis was carried out. It showed the nature and close relationship of the factors studied. Figure 1 presents the graphs showing that with an increase in the HTC values from 0.3 to 1.0, the yield increases in the clusters "semi-dry", "very arid", "arid", "semi-arid" and "subhumid", with the determination coefficients $R^2$ = 0.62–0.78. In the "humid" cluster, with an increase in the HTC values from 1.1 to 1.7, a reduction in yields was noted ($R^2$ = 0.66–0.78) (Figure 1).

The analysis of the dependence of the silage maize on the HTC by the plots with natural fertility and the application of increased fertilizer rates revealed a similar behavior (Table 5).

**Table 5.** Dependence of the silage maize productivity (Y) from the hydrothermal coefficient value (X) with different fertilization levels and tillage methods, 1991–2020.

| Cluster Characteristics According to the HTC | Parameters | Fertilizer Application Level | | | |
| --- | --- | --- | --- | --- | --- |
| | | «0» | «2» | «0» | «2» |
| | | Chisel Method | | Moldboard Method | |
| Semi-dry | Equation | $y = 1605x^2 - 913.35x + 132.61$ | $y = 1511.7x^2 - 861.61x + 128.88$ | $y = 1181.5x^2 - 660.13x + 96.61$ | $y = 1068.2x^2 - 588.76x + 88.69$ |
| | $R^2$ | 0.72 | 0.76 | 0.73 | 0.76 |
| Very arid | Equation | $y = 194.78x^2 - 165.56x + 47.96$ | $y = 180.81x^2 - 95.094x + 23.044$ | $y = 15.809x^2 + 8.5342x + 5.7925$ | $y = 522.21x^2 - 433.07x + 104.79$ |
| | $R^2$ | 0.99 | 0.95 | 0.95 | 0.91 |
| Arid | Equation | $y = -210.65x^2 + 328.2x - 108.63$ | $y = 528.33x^2 - 690.05x + 246.84$ | $y = -196.46x^2 + 311.75x - 104.5$ | $y = 443.17x^2 - 566.26x + 200.65$ |
| | $R^2$ | 0.94 | 0.75 | 0.92 | 0.75 |
| Semi-arid | Equation | $y = 578.64x^2 - 892.04x + 352.15$ | $y = 628.07x^2 - 948.6x + 371.01$ | $y = 517.8x^2 - 787.48x + 309.69$ | $y = 613.81x^2 - 918.64x + 356.69$ |

**Table 5.** *Cont.*

| Cluster Characteristics According to the HTC | Parameters | Fertilizer Application Level | | | |
| --- | --- | --- | --- | --- | --- |
| | | «0» | «2» | «0» | «2» |
| | | Chisel Method | | Moldboard Method | |
| | $R^2$ | 0.88 | 0.89 | 0.82 | 0.90 |
| Subhumid | Equation | $y = -596.59x^2 + 1338.1x - 723.69$ | $y = -328.07x^2 + 767.51x - 410.23$ | $y = -585.65x^2 + 1280.6x - 671.09$ | $y = -456.48x^2 + 1042.3x - 555.62$ |
| | $R^2$ | 0.70 | 0.96 | 0.82 | 0.98 |
| Humid | Equation | $y = -137.86x^2 + 424.39x\ v\ 285.17$ | $y = 4.559x^2 + 3.2645x + 27.523$ | $y = = -165.27x^2 + 511.05x - 351.59$ | $y = -149.2x^2 + 456.2x - 295.4$ |
| | $R^2$ | 0.65 | 0.80 | 0.76 | 0.96 |

Note: "0"—control plot without fertilizers; "2"—FYM 8 t + $N_{120}P_{90}K_{120}$.

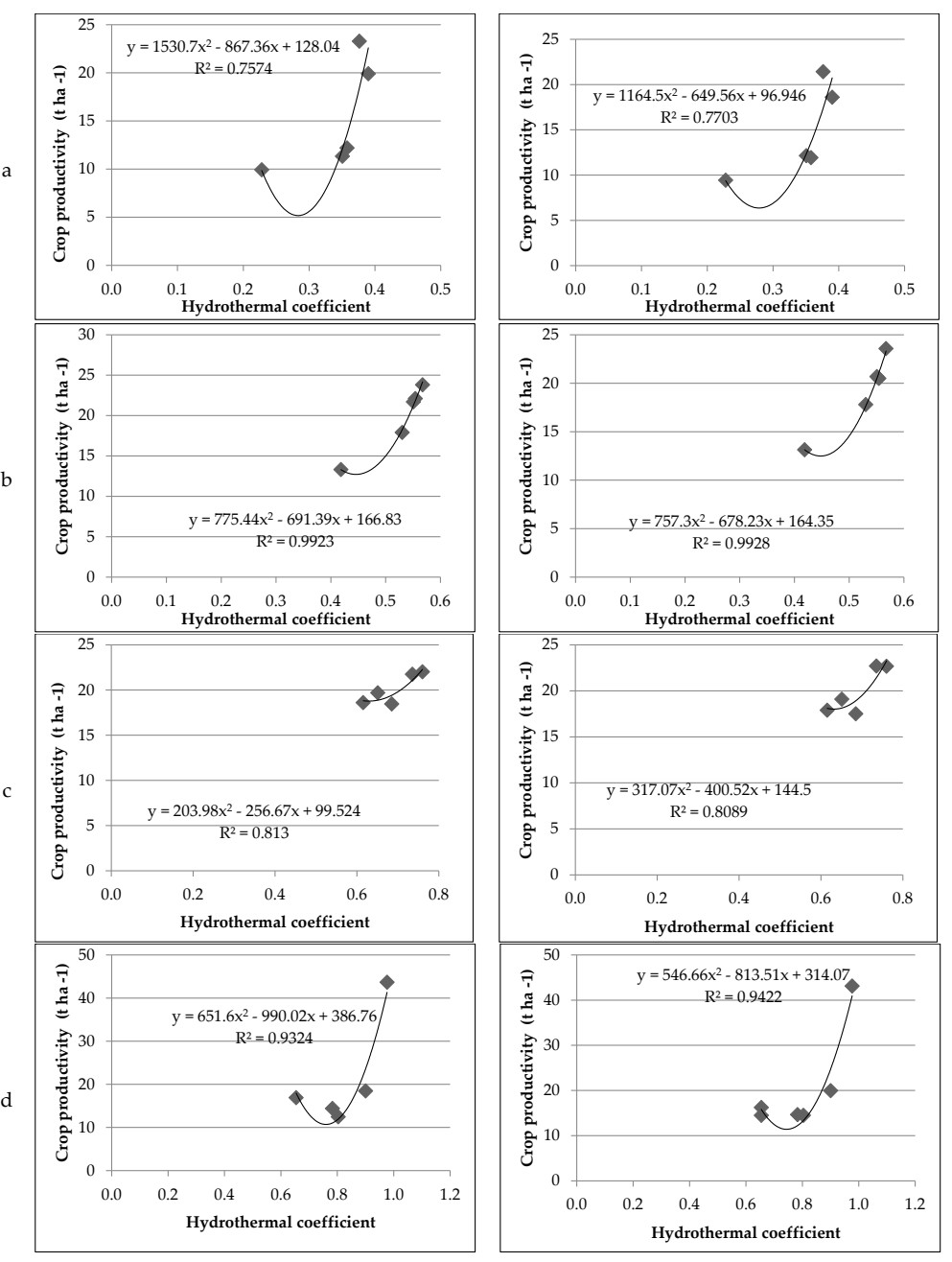

**Figure 1.** *Cont.*

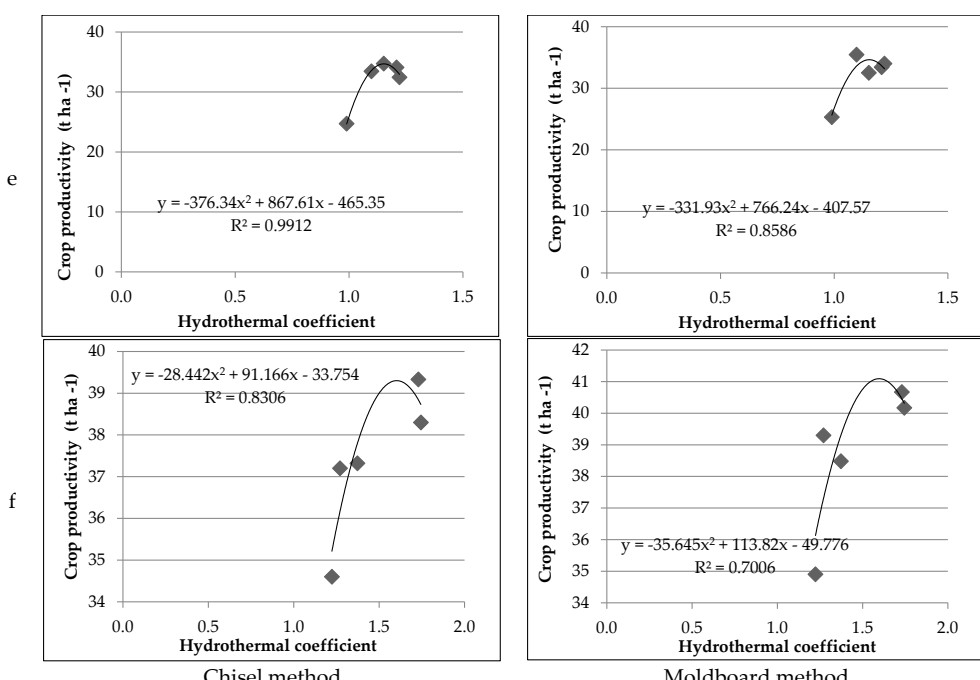

**Figure 1.** Dependence of the silage maize productivity (Y) on the hydrothermal coefficient value (X) with different tillage methods (chisel and moldboard) and an average fertilization level ("1"—FYM 5 t, $N_{80}P_{60}K_{80}$), in various clusters: "semi-dry" (**a**); "very arid" (**b**); "arid" (**c**); "semi-arid" (**d**); "subhumid" (**e**); "humid" (**f**), 1991–2020.

In the years of different meteorological conditions, varying yields were obtained. Grouping all the years of the study into different clusters made it possible to compare the yield with the long-term average annual yield. In the "semi-dry cluster", all variants of fertilization and tillage gave a lower yield by 6.6–7.3 t ha$^{-1}$ (SEM = ±2.3–2.7) compared to the long-term average annual values (33.4–36.3%) (Figure 2).

In the clusters "very arid" and "arid", with an increase of the HTC up to 0.6–0.8 the difference between the actual and long-term average annual yield decreased to −2.7–5.4 t ha$^{-1}$. In the "semi-arid" cluster, the yield was at the level of the long-term average annual values (−1.5–2.8 t ha$^{-1}$), but this difference was statistically insignificant (SEM = ±4.7–4.8). In the "subhumid cluster", a tendency towards a yield increase from 2.7–4.8 t ha$^{-1}$ (SEM = ±1.9–2.0) was noted with the natural fertility level to 8.5–8.7 t ha$^{-1}$ (SEM = ±1.8) with the application of increased fertilizers rates. In the "humid cluster", a statistically significant yield increase was obtained compared to the long-term average annual values: with the natural fertility level it was 14.0–14.1 t ha$^{-1}$ (SEM = ±2.3–2.7), and with an increased fertilization level it was 14.9–17.6 t ha$^{-1}$ (SEM = ±1.8–2.1).

To identify the factors influencing the silage maize, a correlation analysis was carried out with the following parameters: the amount of precipitation in millimeters for the period with temperatures above +10 °C; the sum of air temperatures in degrees Celsius (°C); hydrothermal coefficient.

As a result of the analysis, close correlations between the yield and the sum of precipitation were revealed. In the range of r = 0.33–0.98 and with the HTC in the range of r = 0.61–0.93, higher values of the correlation coefficient were found on the plots with moldboard tillage with an increased fertilizer level (Table 6).

The correlation of corn yield with the sum of the temperatures is most often weakly negative—mostly less than r = 0.5, which is explained by the stronger influence of the moisture factor. In clusters with an increased sum of precipitation, there is a tendency towards a weakening of the close relationship with productivity. The correlation changes from positive to negative and the values of the correlation coefficient decrease. The closest relationship between the silage maize yield and the meteorological factors was noted in the

"arid cluster", where for the sum of temperatures it was r = −0.45–0.64 and for the sum of precipitation it was r = 0.94–0.98, and for r = HTC—0.80–0.93.

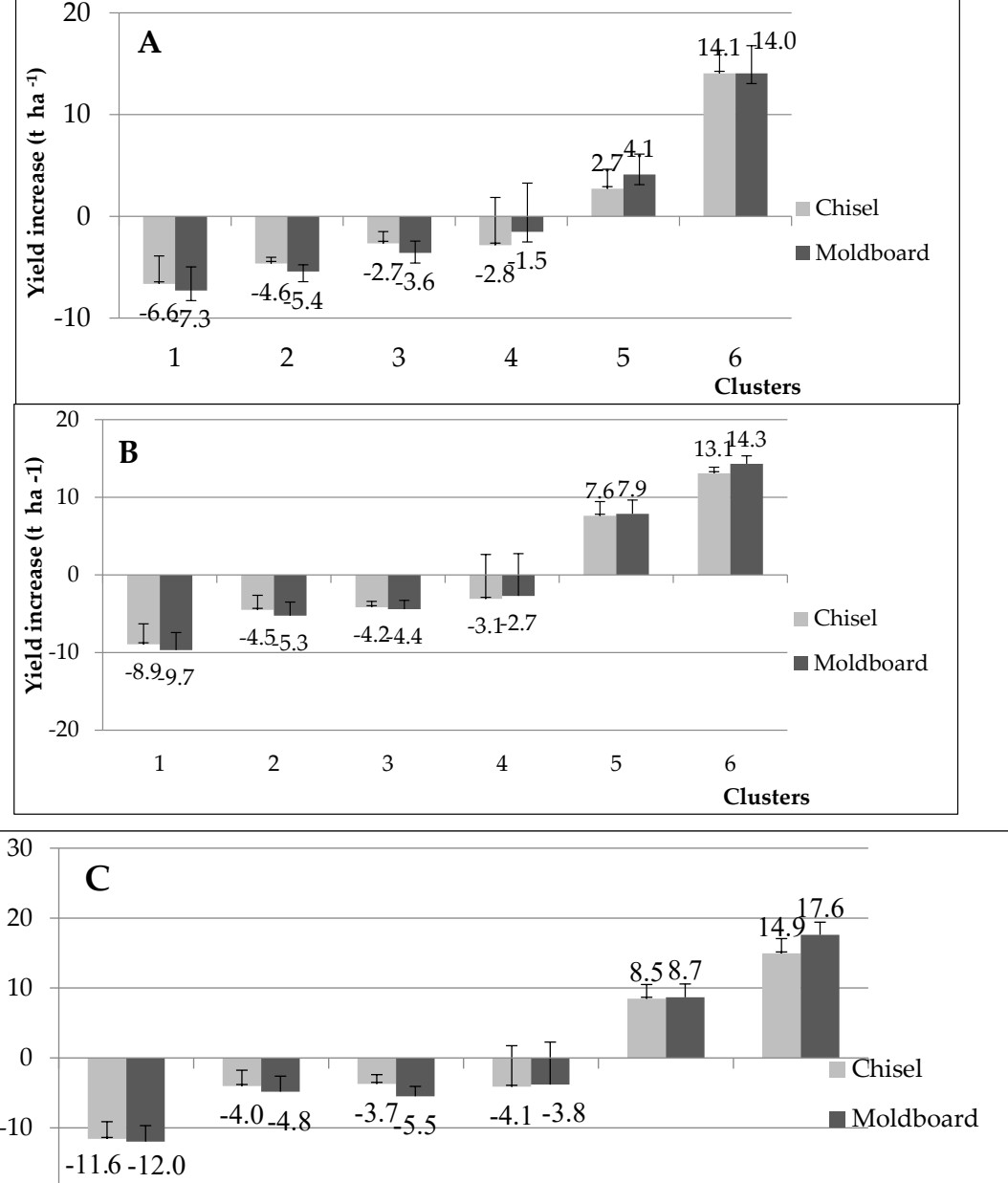

**Figure 2.** Change in the silage maize yield (t ha$^{-1}$) in various clusters: "semi-dry" (1); "very arid" (2); "arid" (3); "semi-arid" (4); "subhumid" (5); and "humid" (6) compared to the long-term average annual values. Standard error (SE) is expressed by error bars. Fertilizer application levels: (**A**)—control plot without fertilizers; (**B**)—FYM 5 t + $N_{80}P_{60}K_{80}$; (**C**)—FYM 8 t + $N_{120}P_{90}K_{120}$, 1991–2020.

A close relationship with the value of the correlation coefficient r = 0.88–0.93 was found between the yield increase and the HTC value in the "arid cluster" with chisel tillage. In the "humid cluster", moldboard tillage had an advantage (r = 0.70–0.85).

**Table 6.** Pearson's correlation coefficients of yield (t ha$^{-1}$) and yield increase (t ha$^{-1}$) depending on basic tillage method and fertilization level in various clusters grouped according to the meteorological indicators of the corn growing season, 1991–2020.

| Clusters | Meteorological Indicator | Productivity, t ha$^{-1}$ | | | | | | Yield Increase, t ha$^{-1}$ | | | |
| | | «0» | | «1» | | «2» | | «1» | | «2» | |
| | | Chisel | Mold-Board | Chisel | Mold-Board | Chisel | Mold-Board | Chisel | Mold-Board | Chisel | Mold-Board |
|---|---|---|---|---|---|---|---|---|---|---|---|
| Semi-dry | P | 0.69 | 0.78 | 0.69 | 0.75 | 0.68 | 0.78 | −0.17 | −0.28 | −0.46 | −0.13 |
| | ΣT | −0.21 | −0.22 | −0.28 | −0.32 | −0.26 | −0.33 | −0.48 | −0.51 | −0.10 | −0.79 |
| | HTC | 0.63 | 0.71 | 0.67 | 0.73 | 0.65 | 0.75 | −0.44 | −0.52 | −0.32 | 0.25 |
| Very arid | P | −0.21 | −0.22 | −0.28 | −0.32 | −0.26 | −0.33 | −0.48 | −0.51 | −0.10 | −0.79 |
| | ΣT | 0.82 | 0.82 | 0.68 | 0.67 | 0.82 | 0.75 | 0.59 | 0.53 | 0.81 | 0.71 |
| | HTC | −0.45 | −0.45 | −0.64 | −0.64 | −0.46 | −0.50 | −0.71 | −0.70 | −0.46 | −0.52 |
| Arid | P | 0.98 | 0.98 | 0.95 | 0.95 | 0.98 | 0.94 | 0.91 | 0.86 | 0.98 | 0.91 |
| | ΣT | −0.45 | −0.45 | −0.64 | −0.64 | −0.46 | −0.50 | −0.71 | −0.70 | −0.46 | −0.52 |
| | HTC | 0.91 | 0.93 | 0.89 | 0.89 | 0.80 | 0.87 | −0.88 | −0.93 | −0.89 | −0.91 |
| Semi-arid | P | 0.44 | 0.42 | 0.51 | 0.51 | 0.47 | 0.47 | 0.62 | 0.94 | 0.76 | 0.94 |
| | ΣT | 0.92 | 0.94 | 0.86 | 0.85 | 0.76 | 0.84 | −0.61 | −0.19 | −0.08 | 0.12 |
| | HTC | 0.61 | 0.51 | 0.54 | 0.54 | 0.47 | 0.49 | 0.60 | 0.94 | 0.72 | 0.93 |
| Subhu-mid | P | 0.33 | 0.32 | 0.44 | 0.48 | 0.42 | 0.45 | 0.77 | 0.89 | 0.64 | 0.80 |
| | ΣT | −0.23 | −0.30 | −0.15 | −0.13 | −0.20 | −0.18 | 0.18 | 0.63 | 0.78 | 0.25 |
| | HTC | 0.61 | 0.66 | 0.70 | 0.75 | 0.71 | 0.75 | 0.93 | 0.78 | 0.92 | 0.93 |
| Humid | P | −0.23 | −0.30 | −0.15 | −0.13 | −0.20 | −0.18 | 0.18 | 0.72 | 0.7 | 0.85 |
| | ΣT | 0.34 | 0.26 | 0.93 | 0.98 | 0.98 | 0.98 | 0.48 | 0.70 | 0.57 | 0.80 |
| | HTC | 0.86 | 0.90 | 0.13 | 0.27 | 0.11 | 0.33 | −0.64 | −0.71 | −0.59 | −0.83 |
| Long-term average annual values | P | −0.21 | −0.32 | −0.80 | −0.76 | −0.87 | −0.72 | 0.86 | 0.92 | 0.91 | 0.91 |
| | ΣT | 0.86 | 0.90 | 0.13 | 0.28 | 0.12 | 0.34 | −0.63 | −0.60 | −0.58 | −0.53 |
| | HTC | 0.38 | 0.49 | 0.75 | 0.64 | 0.91 | 0.48 | −0.17 | −0.37 | 0.42 | −0.26 |

Note: "0"—control plot without fertilizers; "1"—FYM 5 t + N80P60K80; "2"—FYM 8 t + N120P90K120. P—the sum of precipitation in millimeters for the period with temperatures above +10 °C; ΣT—the sum of air temperatures in degrees Celsius (°C) for the same period; HTC—hydrothermal coefficient.

### 3.3.2. Influence of the Tillage Method on the Silage Maize

On average, over the 30-year observation period with chisel tillage, the yield was lower by 0.1–0.3 t ha$^{-1}$ compared to moldboard tillage (SEM = ±0.16–0.24) (Figure 3).

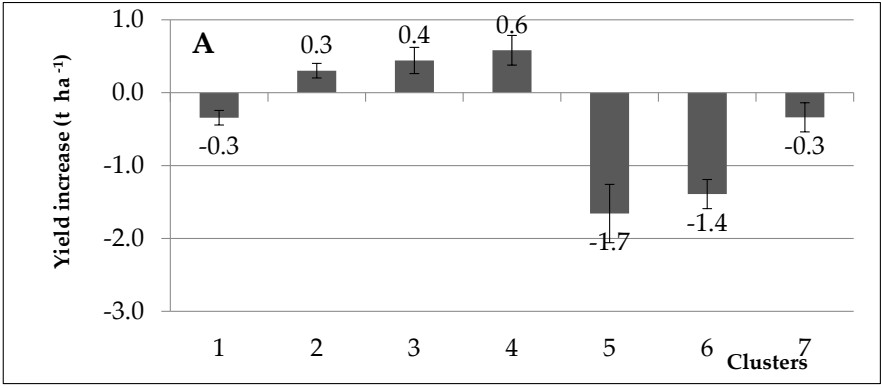

**Figure 3.** *Cont*.

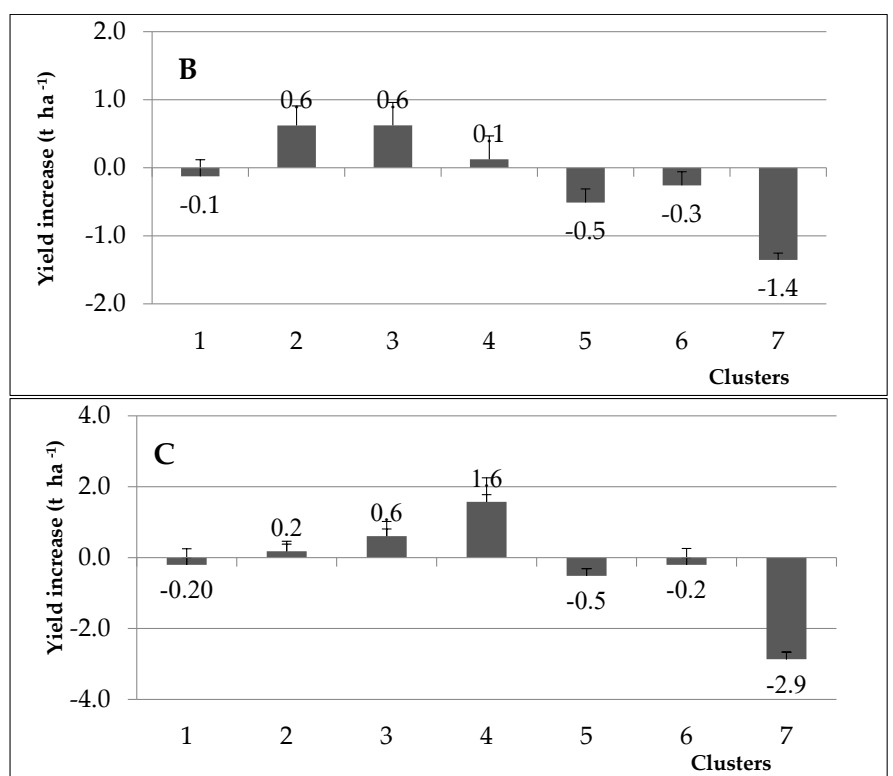

**Figure 3.** Change in the silage maize yield (t ha$^{-1}$) of long-term average annual values (1) in different clusters: "semi-dry" (2); "very arid" (3); "arid" (4); "semi-arid" (5); "subhumid" (6); "humid" (7) with chisel tillage compared to a control method (moldboard tillage was taken as a control method). Fertilizer application levels: (**A**)—control plot without fertilizers; (**B**)—FYM 5 t + N$_{80}$P$_{60}$K$_{80}$; (**C**)—FYM 8 t + N$_{120}$P$_{90}$K$_{120}$, 1991–2020.

In the clusters "semi-dry", "very arid" and "arid", chisel tillage had an advantage over the moldboard method. The difference in the yield was 0.2–1.6 t ha$^{-1}$ (SEM = ± 0.08−0.27). In the clusters "semi-arid", "subhumid" and "humid", the silage maize was higher with moldboard tillage than with chisel method by 0.3–2.9 t ha$^{-1}$ (SEM = ± 0.87−0.61).

3.3.3. Return of Fertilizers on Silage Maize

On average, over the years of the study, the yield increase was 4.3–4.5 t ha$^{-1}$ with the application of an average rate of fertilizers. The application of a higher rate of fertilizers increased the yield up to 7.5–7.7 t ha$^{-1}$. In the clusters "semi-dry", "very arid" and "arid" (HTC 0.3–0.7), the yield increase with the application of an average fertilizer rate was in some cases half of the long-term average annual values and amounted to 1.9–4.7 t ha$^{-1}$. An increase in the fertilizer rate by 1.5 times contributed to the production of extra products by up to 2.7–8.3 t ha$^{-1}$ compared to the long-term average values. In the clusters "semi-arid", "subhumid" and "humid" (HTC 0.9–1.5), the yield increase with the application of an average fertilizer rate increased to 3.4–9.4 t ha$^{-1}$. An increase in the fertilizer rate by 1.5 times in the same clusters made it possible to obtain up to 5.3–13.4 t ha$^{-1}$ of extra products (Table 7).

The efficiency of fertilization was assessed by their return on a yield increase in kilograms. On average, over the years of the study and when applying an average and increased fertilizers rate, the return was 15.0–15.7 kg/kg, and 17.3–17.6 kg/kg with the advantage of chisel tillage. It was found that in arid conditions (HTC = 0.3), the highest return on a yield increase (7.7 kg/kg) was observed in the plot with chisel tillage and an average fertilizers rate, which is 19.0% more as compared to the increased fertilizers rate. In other clusters, a higher return was noted with the application of an increased fertilizers rate. In the clusters "very arid", "arid" and "semi-arid" (HTC 0.5–0.9), the yield increase when

applying a higher fertilizers rate was slightly higher (6.5–44.4%) as compared to applying an average rate. In the "subhumid" and "humid" clusters (HTC 1.2–1.5), the difference between the application of an average and increased fertilizers rate was more than 58–59% in favor of the increased rate, which confirms the more efficient use of fertilizers with good water availability.

**Table 7.** Return of fertilizers on silage maize, depending on the level of fertilizer application and tillage method, 1991–2020.

| Cluster Characteristics According to the HTC | Tillage | Yield Increase, t ha$^{-1}$ | | Return of Fertilizers on a Yield Increase, kg/kg | |
|---|---|---|---|---|---|
| | | «1» | «2» | «1» | «2» |
| Semi-dry | Chisel | 2.2 ± 0.32 | 2.7 ± 0.45 | 7.7 ± 1.10 | 6.3 ± 1.02 |
| | Moldboard | 1.9 ± 0.40 | 2.9 ± 0.30 | 6.6 ± 1.40 | 6.5 ± 0.68 |
| Very arid | Chisel | 4.7 ± 1.27 | 8.3 ± 1.56 * | 16.2 ± 4.42 | 19.0 ± 3.55 |
| | Moldboard | 4.5 ± 1.18 | 8.1 ± 1.45 * | 15.6 ± 4.10 | 18.6 ± 3.30 |
| Arid | Chisel | 3.0 ± 0.54 | 6.6 ± 0.85 * | 10.5 ± 1.87 | 15.2 ± 1.95 |
| | Moldboard | 3.5 ± 0.66 | 5.7 ± 0.87 ** | 12.1 ± 2.30 | 12.9 ± 1.98 |
| Semi-arid | Chisel | 4.3 ± 1.08 | 6.4 ± 1.11 * | 14.9 ± 3.74 | 14.6 ± 2.54 |
| | Moldboard | 3.1 ± 1.15 | 5.3 ± 1.32 * | 10.9 ± 4.01 | 12.0 ± 3.02 |
| Subhumid | Chisel | 9.4 ± 0.71 | 13.4 ± 1.92 * | 32.7 ± 2.47 | 30.7 ± 4.38 * |
| | Moldboard | 8.3 ± 0.78 | 12.2 ± 2.01 * | 28.8 ± 2.72 | 28.0 ± 4.58 * |
| Humid | Chisel | 3.6 ± 0.86 | 8.6 ± 2.30 * | 12.4 ± 2.99 | 19.6 ± 5.25 * |
| | Moldboard | 4.6 ± 1.31 | 11.1 ± 2.20 * | 15.9 ± 4.55 | 25.4 ± 5.02 * |
| Long-term average annual values | Chisel | 4.5 ± 0.37 | 7.7 ± 0.72 *** | 15.7 ± 1.29 | 17.6 ± 1.65 |
| | Moldboard | 4.3 ± 0.39 | 7.5 ± 0.73 *** | 15.0 ± 1.37 | 17.3 ± 1.67 |

Note: "1"—FYM 5 t + N80P60K80; "2"—FYM 8 t + N120P90K120. Abbreviations: na, not available; ns, not significant. * $p < 0.05$, ** $p < 0.01$, *** $p < 0.001$.

Under drought conditions ("semi-dry cluster"), no dependence between the yield increase and the HTC was found ($R^2$ = 0.02–0.19). In other years, there was a close dependence confirmed by the coefficient of determination $R^2$ = 0.5–0.9 (Table 8).

**Table 8.** Dependence of the yield increase of silage maize (Y) on the HTC value (X) with different levels of fertilizer application and tillage methods, 1991–2020.

| Clusters | Parameters | Fertilizers Application Level | | | |
|---|---|---|---|---|---|
| | | «1» | «2» | «1» | «2» |
| | | Chisel | | Moldboard | |
| Semi-dry | Equation | $y = -74.331x^2 + 45.99x - 4.5659$ | $y = -93.354x^2 + 51.737x - 3.7329$ | $y = = -17.027x^2 + 10.576x + 0.336$ | $y = -113.33x^2 + 71.375x - 7.91$ |
| | $R^2$ | 0.06 | 0.14 | 0.02 | 0.19 |
| Very arid | Equation | $y = 747.73x^2 - 686.41x + 156.81$ | $y = 153.1x^2 - 90.108x + 13.026$ | $Y = 741.49x^2 - 686.77x + 158.56$ | $y = 506.4x^2 - 441.61x + 99.001$ |
| | $R^2$ | 0.99 | 0.96 | 0.94 | 0.88 |
| Arid | Equation | $y = 438.2x^2 - 619.89x + 220.87$ | $y = 762.55x^2 - 1053.3x + 368.19$ | $y = 513.54x^2 - 712.27x + 249$ | $y = 762.55x^2 - 1053.3x + 368.19$ |
| | $R^2$ | 0.80 | 0.84 | 0.72 | 0.84 |

**Table 8.** *Cont.*

| Clusters | Parameters | Fertilizers Application Level | | | |
|---|---|---|---|---|---|
| | | «1» | «2» | «1» | «2» |
| | | Chisel | | Moldboard | |
| Semi-arid | Equation | $y = -72.96x^2 - 97.986x + 34.612$ | $y = 49.432x^2 - 56.565x + 18.86$ | $= 56.803x^2 - 75.389x + 26.007$ | $y = 96.009x^2 - 131.16x + 47.003$ |
| | $R^2$ | 0.98 | 0.89 | 0.67 | 0.99 |
| Subhumid | Equation | $y = 171.41x^2 - 334.66x + 167.27$ | $y = 219.68x^2 - 434.77x + 222.39$ | $y = 253.72x^2 - 514.39x + 263.52$ | $y = 129.18x^2 - 238.32x + 115.46$ |
| | $R^2$ | 0.79 | 0.90 | 0.95 | 0.85 |
| Humid | Equation | $y = 121.88x^2 - 367.5x + 274.81$ | $y = 150.72x^2 - 443.95x + 328.27$ | $y = 145.37x^2 - 440.53x + 331.37$ | $y = 150.72x^2 - 443.95x + 328.27$ |
| | $R^2$ | 0.62 | 0.50 | 0.86 | 0.70 |

Note: "1"—FYM 5 t + $N_{80}P_{60}K_{80}$; "2"—FYM 8 t + $N_{120}P_{90}K_{120}$.

## 4. Discussion

Climate change significantly affects silage maize. During the study, more than half of the years (69%) were severely arid and arid (HTC = 0.3–0.9). Drought stress and insufficient water availability, as well as relatively low air temperature and excessive precipitation, influenced the content of available moisture in the soil. Under these conditions, the corn yield varied within a wide range (12.8–45.2 t ha$^{-1}$). Temperature and precipitation changes can have both positive and negative impacts on corn yields [24].

The most important period for corn development is May. In this month, meteorological conditions vary widely (HTC 0.3–1.7). Sprouts and future harvests depend on water availability. With abundant rainfall and low temperatures, the sprouts are even. However, along with corn, weeds appear. Their germination temperatures are sufficiently lower. With excessive air humidity, diseases occur. To protect plants from diseases and weeds, farmers have to invest additional costs to preserve the harvest [25,26]. Soil drought, on the contrary, leads to thinned sprouts and a reduction in yield [27,28]. Plants have to adapt to drought [29].

Corn belongs to moisture-loving crops; therefore, a significant amount of precipitation during the growing season (265 mm) compared to the long-term average annual values (159 mm) and the decrease in the average monthly air temperature during the growing season to 19.8 °C (long-term average annual value—22.1 °C) made it possible to obtain up to 42.4–45.2 t ha$^{-1}$ of silage maize.

In dry periods, the silage maize yield was lower by 1.5–9.7 t ha$^{-1}$ compared to the long-term average annual values; with the HTC increase, this difference decreased. With optimal values of the HTC = 0.9–1.2, the yield increase was 2.7–14.3 t ha$^{-1}$. Organic fertilizers, used with mineral fertilizers, are able to mitigate plant stress during drought and high air temperatures [30]. The use of medium rates of organomineral fertilizer systems increased yields by 10.5–41.9%, and with higher rates by 21.0–59.8% [31,32]. As a result of the analysis of the yield increase in individual clusters in comparison with the long-term average annual values, general tendencies were revealed. In severely arid conditions, moisture-saving chisel tillage had an advantage. The silage maize yield was 0.3–1.2 t ha$^{-1}$ higher under these conditions as compared to the moldboard tillage. The water availability improvement in the growing season (HTC = 1.2–1.5) combined with the optimal tillage method for the given conditions (moldboard tillage) made it possible to obtain 1.5–3.0 times more extra products. A significant yield increase was found only in two clusters: "semi-arid" and "subhumid".

Optimal ratios of temperature and water availability allow for the obtaining of the maximum yield. However, with HTC increases above 1.74, the corn yield decreases. The maximum silage maize in the "subhumid cluster" is noted at values of HTC = 1.1 (39.1 t ha$^{-1}$),

and in the "humid cluster" at values of HTC = 1.4 (49.9 t ha$^{-1}$). A 1.5-fold increase in precipitation and a slight decrease in the average monthly temperature compared to the long-term average annual values led to a reduction in yield.

In the steppe regions of southern Russia, most of the years are severely arid and arid, so farmers are advised to apply chisel tillage for corn. Chisel tillage, in contrast to moldboard, is able to retain and accumulate available moisture due to the creation of a mulching layer on the surface, and the silage maize yield is higher than with moldboard tillage [33,34].

To select the optimal combination of factors affecting the yield and the corn yield increase under various weather conditions, it is necessary to determine the close relationship between them. The correlation ratio shows it. The analysis shows a positive correlation between corn yield and water availability, and between the sum of the temperatures in the growing season and the HTC value, both for individual clusters and between long-term average annual data [35]. The correlation coefficient increases with a higher nutritional level.

One of the important economic characteristics is the efficiency of fertilization. In the arid conditions, silage maize yield and the return on fertilizer were low. A further increase in the fertilizer application rate did not lead to yield increase, and the return on fertilizer was low. Lack of moisture, drought stress, and all these negative environmental conditions prevented the plants from developing properly. The return on fertilizers was higher with chisel tillage than with moldboard method. With an increase in water availability, the role of fertilizers increases. When applying a higher rate of fertilizers, their return increases. In "subhumid" and "humid" clusters, the return with the increased application rate of fertilizers is 1.5–2 times higher than with the average rate.

## 5. Conclusions

In the conditions of the Rostov region, the water supply of the growing season is a decisive factor in the formation of the corn crop. Over 30 years of observation, a strong positive correlation has been revealed between the corn yield and the amount of precipitation and a negative one with the sum of temperatures. In dry conditions, when growing corn for silage, chisel tillage is most effective. In wet years, moldboard processing should be preferred. In dry years, the most effective average application rate of fertilizers (5 t FYM + $N_{80}P_{60}K_{80}$) with the highest return to fertilizers is up to 7.7 kg/kg due to an increase in yield. In the rest of the years, a higher return from fertilizers by harvest was observed on variants with an increased rate of fertilizer (8 t FYM + $N_{120}P_{90}K_{120}$). Thus, our study provides valuable recommendations in fodder production, promotes moisture conservation, preserves soil fertility on slopes when cultivating corn for silage, and will be useful to specialists in improving the efficiency of agricultural production.

**Author Contributions:** Conceptualization, E.G. and O.B.; methodology, I.I.; software, E.N.; validation, S.T. and E.N.; formal analysis, A.M.; resources, E.N.; data curation, A.K.; writing—original draft, E.G.; writing—review and editing, E.G. and A.G.; visualization, S.T. All authors have read and agreed to the published version of the manuscript.

**Funding:** This research received no external funding.

**Data Availability Statement:** Presented data in this study are available on request from the corresponding author.

**Acknowledgments:** The work was carried out within the framework of the State tasks of Federal State Budget Scientific Institution «Federal Rostov Agricultural Research Centre» on topic No. 0710-2019-0026 "Fundamental principles for creating a new generation of arable farming systems and agricultural technologies in order to preserve and reproduce soil fertility, effectively use the natural-resources potential of agricultural landscapes and produce the given quantity and quality of agricultural products".

**Conflicts of Interest:** The authors declare no conflict of interest.

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
