# Peer review of "Influence of Heat Stress and Water Availability on Productivity of Silage Maize (Zea mays L.) under Different Tillage and Fertilizer Management Practices in Rostov Region of Russia"

_agronomy, doi:10.3390/agronomy13020320_

Round 1

Reviewer 1 Report

Authors need to read all suggestions given by reviewers in this attached manuscript PDF. Kindly check comments on yellow-highlighted texts and comment boxes. The authors are requested to include all suggestions before re-submitting the revised version.

Author Response

Correct the title as "Influence of heat stress and water availability on productivity of silage maize (Zea mays L.) under different tillage and fertilizer management practices in Rostov region of Russia"

We thank you and the reviewers for the careful consideration and comments  of our manuscript entitled «Influence of heat stress and water availability on productivity of silage maize (Zea mays L.) under different tillage and fertilizer management practices in Rostov region of Russia » agronomy-2125731. We have revised the paper thoroughly in response to their comments. We have also attached a version that highlighted the changes we have made to the manuscript by yellow color.

Corrected the remark.

experiments

A long-term multifactorial experiment was performed during 1991-2020 in the South of Rostov region, Russia.

temperature should be in average rather than in total or sum.

«… of 265 mm rainfall and 19.8 °C average temperature during the growing season.»

33.8 - 45.2

Corrected the remark.

fertilizers

Replaced «The application of farmyard manure for fertilization…»

What is the further suggestion or recommendation of your findings?

Our study provides valuable recommendations in fodder production, promotes moisture conservation, preserves soil fertility on the slopes when cultivating corn for silage, and will be useful to specialists in improving the efficiency of agricultural production.

L28 Add 'heat stress' as one of the key words.

Calcic Chernozem (Pachic), chisel moldboard, chisel tillage, fertilizers,  heat stress,

Add below information in Introduction section;

1) Current production and productivity of silage maize in Russia and study region.

2) Relate the productivity of silage maize with heat stress and  reduced availability of water ; how do they (heat stress and water shortage) affect the productivity of silage maize.

3) Add information on tillage and fertilizer management practices for cultivating silage maize in the study region.

4) What is the research gap?, what is the justification of your study?. Write the objective of the experiment or study?

Add information with latest references regarding the water stress  on production of silage maize.

The yield of corn for silage in Russia ranges from 50-60 t ha-1, under the arid conditions of the Rostov region, it decreases to 25-45 t ha-1[3].

Contemporary agriculture is in an urgent need to increase the efficiency of mineral fertilizers. Firstly, they should provide plants  with mineral elements in optimal amounts at all stages of development. Secondly, mineral fertilizers should replenish the soil pool of the chemical elements used by the plants for crop production.

In arid conditions, the method of tillage is of great importance. The main advantage of chisel cultivation is the reduction in plowing costs, the prevention of water and wind erosion on flat and sloping areas, the improvement of physical, mechanical and biochemical parameters, the preservation of moisture in soil horizons, as well as the increase in fertility. In erosion-prone agricultural landscapes, soil protection technologies are based on non-moldboard tillage, which reduces soil erosion by 2–4 times in crop rotations with tilled crops [12].

The article presents the results of research of corn cultivation conditions on slope erosion-hazardous lands, which determines the novelty of the work and significantly expands the range of application of the results in practice.

under different tillage and fertilizer management practices

Corrected the remark.« The purpose of our study is to determine the influence of heat stress and water availability on the silage maize in a under different tillage and fertilizer management practices and to find out the effectiveness of technological methods that ensure maximum use of available moisture».

Add information regarding which variety of maize (add its source of origin) was used in this study?

Write the materials and methods in below sub-headings;

2.1. Description of study site:

2.2. Experimental design, treatments and cultural practices:

2.3. Data observation and calculation:

2.4. Data analysis: add statistical analysis also.

Corrected the remark.

check this.

Corrected the remark

47.370022 N, 39.909648 E; 47.36958 N, 39.90929 E.

Name of this hybrid??, Where is its source of origin ?

A mid-season corn hybrid Zernogradsky 282 MV was sown, bred at the Donskoy Agricultural Research Center, Zernograd, Russia.

Provide the name of the experimental design and replication number.

Over thirty years of research, there have been several names. We have given the name currently used.

«The experiment was registered in the Russian Geographical Network of long-term experiments with fertilizers application (certificate №. 169)».

What is its name of this fertilizer ? and where is the source of this fertilizer ?

Phosphorus (ammophos) and potash (potassium chloride) fertilizers were applied before the main tillage, top dressing was carried out with ammonium nitrate in the phase of 2-4 leaves.

It is better to write temperature in average in stead of total or sum.

Corn ripens when the sum of temperatures is set is 3210 – 3400 °C. This is what we have indicated.

Provide information of year. Which year or years?

These data are given from the directory: «Selyaninov, G.T. Methodology for agricultural climate characteristics. World agroclimatic reference book. Leningrad. 1977, 220p». Аksaysky Region

Which version of this Excel was used?

Which mean comparison test was applied?

Two-way analysis of variance (ANOVA) was performed to assess the influence of tillage methods and fertilizers application rates on the productivity silage maize. Fisher (LSD test) was used as a post-hoc test at the 95% level (p 0.05) of significance. Statistical processing and visualization of the obtained results was carried out by the variance analysis [23] using Microsoft Excel 2011 and Statistica 13 (StatSoft, USA).

The authors provided results without mentioning or referring table or figure number in results texts. Add the respective tables or figure in respective place in results.

Corrected the remark.

Refer which Table?

Table 2

Provide information of year. Which year or years?

The studies were carried out at the Federal State Budgetary Scientific Institution "Federal Rostov Agricultural Research Centre" in 1991-2020.

Refer which Table?

Table 2

Add reference for this information.

The data were obtained by us (Table 3).

Provide information of year. Which year or years?.

It is not necessary to write 'na' in superscript in the column in this table.

Provide LSD (0.05) values in the table.

It is better to write fertilization levels i.e., 0, 1, 2 in foot notes of table rather than in table heading.

Note

"0" – сontrol plot without fertilizers ; "1" – FYM 5 t + N80P60K80; "2" – FYM 8 t + N120P90K120

*p < .05, **p < .01, ***p < .001.

Refer which Table?

Table 3

Provide information of year. Which year or years?.

Write 0, 1, 2 in foot notes of this table rather than in table heading.

1991-2020

Corrected the remark

Provide information of year. Which year or years?.

Provide the number of each figure like 1A, 1B, 1C etc

Corrected the remark

Provide information of year. Which year or years?.

Write in footnote of table regarding  what do the 0,  2 indicate for what information?

1991-2020, "0" – сontrol plot without fertilizers ; "2" – FYM 8 t + N120P90K120 Note

"0" – сontrol plot without fertilizers ; "2" – FYM 8 t + N120P90K120

*p < .05, **p < .01, ***p < .001.

Provide information of year. Which year or years?.

Add the name of X-axis in this figure.

What does the error bar( line in top of bars) indicates for what information?

Coefficient of variation of silage maize yield under the influence of tillage and nutritional level, %, 1991-2020.

Tillage, fertilizer application rate

Note

"0" – сontrol plot without fertilizers ; "1" – FYM 5 t + N80P60K80; "2" – FYM 8 t + N120P90K120

Refer which Table or figure?

Figure 2

Provide information of year. Which year or years?.

Write in footnote of this table regarding what do 0,  1, 2 indicate for what information?

The table heading of table 6 is too long, reduce it.

It would be better to provide significant level of correlation values (*, **, *** ..etc)

Which method of correlation (Pearson correlation??) was applied here?

Corrected the remark

Refer which Table?

Table 6.

Refer which Table?

Table 6.

What do the A, B and C (in X-axis) indicate for what information?, Write  the name of the X-axis instead of writing symbol A, B and C.

Corrected the remark

Provide information of year. Which year or years?.

Corrected the remark

Refer which Table or figure?

Figure 3.

Provide information of year. Which year or years?.

It is not necessary to write 'na'  in superscript in the column of this table.

Provide LSD (0.05) values in the table.

Corrected the remark

It is not necessary to write 'na' in column of this table.

Corrected the remark.

Refer which Table?

Table 7

Provide information of year. Which year or years?.

Write in footnote of this  table regarding what does << 1>>,  <<2>> indicate for what information?

Corrected the remark.

Check this

Corrected the remark

Re-write the conclusion.

Do not repeat the result data (numerical values) in conclusion section.

Include main findings of effect of heat stress and water shortage on productivity of silage maize under different tillage and fertilizer management practices.

Provide further recommendation based on your main findings.

In the conditions of the Rostov region, the water supply of the growing season is a decisive factor in the formation of the corn crop: over 30 years of observations, a strong positive correlation has been revealed between the corn yield and the amount of precipitation and a negative one with the sum of temperatures. In dry conditions, when growing corn for silage, chisel tillage is most effective. In wet years, moldboard processing should be preferred. In dry years, the most effective is average application rate of fertilizers (5 t FYM + N80P60K80) with the highest return to fertilizers up to 7.7 kg/kg due to an increase in yield. In the rest of the years, a higher return from fertilizers by harvest was observed on variants with an increased rate of fertilizer (8 t FYM+ N120P90К120). Thus, our study provides valuable recommendations in fodder production, promotes moisture conservation, preserves soil fertility on slopes when cultivating corn for silage, and will be useful to specialists in improving the efficiency of agricultural production.

All references should follow the journal guidelines.

Provide DOI for journal references.

Other changes: We do not list other changes here but have marked them in yellow in the revised manuscript.

Reviewer 2 Report

The paper presents interesting findings relevant to climate change (water and heat stress). However, it requires some effort to bring the paper into proper shape. It requires moderate English edits by an expert on the subject. Also, there are a lot of issues related to the scientific way of presentation. I hope that would be taken care of while going for a scientific English edit. I have pointed out some of the specific errors that authors need to consider while revising the manuscript.

L 3: Italicise the S.N.

L 20: Correct temperature if it is in range 18-20C

L 21: Replace comma (,) with decimal (.)

L 26: Check unit

L 31: Italicize the scientific name

L 41: Here 'problem' is not proper word when talk about solution of problem (increase efficiency). Accordingly, replace with proper word something like 'need' or 'necessity'.

L 42-43: Language issue, both statements are not connected properly.

L 70-72: The statement is very poor: ín climate change'....please improve it.

L 77: Replace 'in' by during before 1991-2000

L 88: Italicize the scientific name of maize

L 93: Remove parenthesis ....FYM

L 98: Remove year after method, not needed in this type of format of citation.

L 107-108: notations differ for different month/ season, so unify them for clear readability to readers

L 115: Remove year

L 141-142: Table 2: I suggest to write  

Precipitation (mm)

Average temperature (oC)  

L 187: lower than what?

L 217:Remove comma (,) after crop productivity from each graph.

2. adjust Equation within the figure scale for better readability.

3. Unify the text size of Axis title (Hydrothermal coefficient)

L 236: It is better to place figure numbering in alphabetical order (A, B, C for three figures) at top-left corner in Bold.

Delete comma (,) after yield increase in Y axis title; slight decrease in font size of data points at each bar.

L 268: Remove comma (,) after productivity and insert unit in parenthesis in each table and figure for uniformity.

L 284: Same as above...

L 285: Follow suggestions for the Axis title as suggested earlier.

L 350: Replace normal dash (-) with em-dash between 42.4 and 45.2, also at other places where range is given; 

and correct -1 in superscript

Author Response

L 3: Italicise the S.N.

We thank you and the reviewers for the careful consideration and comments  of our manuscript entitled «Influence of heat stress and water availability on productivity of silage maize (Zea mays L.) under different tillage and fertilizer management practices in Rostov region of Russia » agronomy-2125731. We have revised the paper thoroughly in response to their comments. We have also attached a version that highlighted the changes we have made to the manuscript by yellow color.

L 20: Correct temperature if it is in range 18-20C

The optimum conditions for silage maize development were found under the combination of 265 mm rainfall and 19.8 °C average temperature during the growing season.

L 21: Replace comma (,) with decimal (.)

33.8-45.2 t ha-1

L 26: Check unit

Our study provides valuable recommendations in fodder production, promotes moisture conservation, preserves soil fertility on the slopes when cultivating corn for silage, and will be useful to specialists in improving the efficiency of agricultural production.

L 31: Italicize the scientific name

(Zea mаys L)

L 41: Here 'problem' is not proper word when talk about solution of problem (increase efficiency). Accordingly, replace with proper word something like 'need' or 'necessity'.

Contemporary agriculture is in an urgent need to increase the efficiency of mineral fertilizers.

L 42-43: Language issue, both statements are not connected properly.

Firstly, they should provide plants  with mineral elements in optimal amounts at all stages of development. Secondly, mineral fertilizers should replenish the soil pool of the chemical elements used by the plants for crop production.

L 70-72: The statement is very poor: ín climate change'....please improve it.

The purpose of our study is to determine the influence of heat stress and water availability on the silage maize yield under different tillage and fertilization practices and to find out the effectiveness of technological methods that ensure maximum use of available moisture.

L 77: Replace 'in' by during before 1991-2000

The studies were carried out in 1991-2020 at the Federal State Budgetary Scientific Institution "Federal Rostov Agricultural Research Centre" in a long-term multifactorial experiment.

L 88: Italicize the scientific name of maize

Corrected the remark.

L 93: Remove parenthesis ....FYM

Corrected the remark.

L 98: Remove year after method, not needed in this type of format of citation.

Corrected the remark.

L 107-108: notations differ for different month/ season, so unify them for clear readability to readers

The average annual temperature is 8.8°С, the average temperature in January (-6.6°С), minimum temperature in winter – minus 41°С, in July + 23°С maximum temperature in summer – up to + 40°С.

L 115: Remove year

…. «was calculated using G.T. Selyaninov’s method».

L 141-142: Table 2: I suggest to write  

Precipitation (mm)

Average temperature (oC)  

Corrected the remark.

L 187: lower than what?

In the clusters with HTC values from 0.3 ("semi-dry cluster") up to 0.5 ("very arid cluster"), the yield was lower than long-term average by 4.0-11.9 t ha-1. In the "arid cluster" (HTC = 0.7), the decrease was slighter – 2.7-5.5 t ha-1.

L 217:Remove comma (,) after crop productivity from each graph.

2. adjust Equation within the figure scale for better readability.

3. Unify the text size of Axis title (Hydrothermal coefficient)

Corrected the remark.

L 236: It is better to place figure numbering in alphabetical order (A, B, C for three figures) at top-left corner in Bold.

Delete comma (,) after yield increase in Y axis title; slight decrease in font size of data points at each bar.

Corrected the remark.

L 268: Remove comma (,) after productivity and insert unit in parenthesis in each table and figure for uniformity.

Corrected the remark.

L 284: Same as above...

Corrected the remark.

L 285: Follow suggestions for the Axis title as suggested earlier.

Corrected the remark.

L 350: Replace normal dash (-) with em-dash between 42.4 and 45.2, also at other places where range is given; and correct -1 in superscript.

Corrected the remark.

Reviewer 3 Report

Please open the attached file

Author Response

1. Keywords should be in alphabetical order and should not duplicate words appearing in the title of the manuscript.

We thank you and the reviewers for the careful consideration and comments  of our manuscript entitled «Influence of heat stress and water availability on productivity of silage maize (Zea mays L.) under different tillage and fertilizer management practices in Rostov region of Russia » agronomy-2125731. We have revised the paper thoroughly in response to their comments. We have also attached a version that highlighted the changes we have made to the manuscript by yellow color.

Calcic Chernozem (Pachic), chisel moldboard, chisel tillage, fertilizers,  heat stress, hydrothermal coefficient, yield.

2. In Abstract section, give a logical reason for selecting the current strategy, i.e., heat stress and water availability on silage maize (Zea mays L.) productivity.

Thanks for your helpful suggestion. We have added a contextual sentence about the motivation of this the study in the revised the manuscript.

A long-term multifactorial experiment is located in a zone of high heat supply and insufficient moisture, so the latter factor limits the yield of corn and dictates the need to find solutions to increase its environmental sustainability in extreme weather conditions.

3. In the introduction section, the author should provide a novelty statement at the end. What new things have authors done or correlated in this research compared to old ones?

The article presents the results of research of corn cultivation conditions on slope erosion-hazardous lands, which determines the novelty of the work and significantly expands the range of application of the results in practice.

4. The authors should follow the title in the introduction section, Influence of heat stress and water availability on silage maize (Zea mays L.) productivity in a changing climate of the Rostov region (Russia)". Do you consider the topic original or relevant in the field? Does it address a specific gap in the field?,

Our study provides valuable recommendations in fodder production, promotes moisture conservation, preserves soil fertility on the slopes when cultivating corn for silage, and will be useful to specialists in improving the efficiency of agricultural production.

5 .The conclusions is so much descriptive. Please provide a conclusive conclusion, Add the targeted beneficiary audience who will get benefit from this research. Also, give clear-cut recommendations Give future prospective regarding this research.

In the conditions of the Rostov region, the water supply of the growing season is a decisive factor in the formation of the corn crop: over 30 years of observations, a strong positive correlation has been revealed between the corn yield and the amount of precipitation and a negative one with the sum of temperatures. In dry conditions, when growing corn for silage, chisel tillage is most effective. In wet years, moldboard processing should be preferred. In dry years, the most effective is average application rate of fertilizers (5 t FYM + N80P60K80) with the highest return to fertilizers up to 7.7 kg/kg due to an increase in yield. In the rest of the years, a higher return from fertilizers by harvest was observed on variants with an increased rate of fertilizer (8 t FYM+ N120P90К120). Thus, our study provides valuable recommendations in fodder production, promotes moisture conservation, preserves soil fertility on slopes when cultivating corn for silage, and will be useful to specialists in improving the efficiency of agricultural production.

7- Mistakes in English language, So the English language of manuscript should be revised by professional in English language

8- the format of table should be layout in the format of MDPI

In all, the comments of editor and reviewers are all helpful to us, and we have revised the paper point by point.

9- In the statistical analysis section, the author should mentioned the type of ANOVA (one way or two way), also the name of program has been used and number of biological samples has been taken

Thanks for constructive comments and suggestions.

Two-way analysis of variance (ANOVA) was performed to assess the influence of tillage methods and fertilizers application rates on the productivity silage maize. Fisher (LSD test) was used as a post-hoc test at the 95% level (p 0.05) of significance. Statistical processing and visualization of the obtained results was carried out by the variance analysis [23] using Microsoft Excel 2011 and Statistica 13 (StatSoft, USA).

The number of biological samples is given in the text (N=5).

10- The authors should added the significant letter above the columns

Other changes: We do not list other changes here but have marked them in yellow in the revised manuscript.

Round 2

Reviewer 3 Report

The authors mad all the correction well and I recommend publishing the manuscript.